# The Hippocampus in Pigeons Contributes to the Model-Based Valuation and the Relationship between Temporal Context States

**DOI:** 10.3390/ani14030431

**Published:** 2024-01-29

**Authors:** Lifang Yang, Fuli Jin, Long Yang, Jiajia Li, Zhihui Li, Mengmeng Li, Zhigang Shang

**Affiliations:** 1School of Electrical and Information Engineering, Zhengzhou University, Zhengzhou 450001, China; flyer1014@163.com (L.Y.); ljinfuli0313@126.com (F.J.); longyang_zzu@163.com (L.Y.); jiajia_li@gs.zzu.edu.cn (J.L.); lizhrain@zzu.edu.cn (Z.L.); 2Henan Key Laboratory of Brain Science and Brain-Computer Interface Technology, Zhengzhou 450001, China; 3Institute of Medical Engineering Technology and Data Mining, Zhengzhou University, Zhengzhou 450001, China

**Keywords:** pigeon, hippocampus, model-based valuation, representation of relationships, local field potentials

## Abstract

**Simple Summary:**

Model-based decision-making guides organism behavior by representing the relationships between states. Previous studies have shown that the mammalian hippocampus (Hp) plays a key role in model-based learning. However, the hippocampal neural mechanisms of birds for model-based learning are largely unknown. We trained pigeons to perform a two-step task. Using a combination of neural analysis and computational modeling, we show that the pigeons use model-based inferences to learn multi-step tasks, and multiple LFP frequency bands collaboratively contribute to model-based learning. Specifically, the high-frequency (12–100 Hz) oscillations represent model-based valuations, while the low-frequency (1–12 Hz) neural similarity is influenced by the relationship between temporal context states. These findings expand the understanding of the hippocampus’ role in avian model-based learning.

**Abstract:**

Model-based decision-making guides organism behavior by the representation of the relationships between different states. Previous studies have shown that the mammalian hippocampus (Hp) plays a key role in learning the structure of relationships among experiences. However, the hippocampal neural mechanisms of birds for model-based learning have rarely been reported. Here, we trained six pigeons to perform a two-step task and explore whether their Hp contributes to model-based learning. Behavioral performance and hippocampal multi-channel local field potentials (LFPs) were recorded during the task. We estimated the subjective values using a reinforcement learning model dynamically fitted to the pigeon’s choice of behavior. The results show that the model-based learner can capture the behavioral choices of pigeons well throughout the learning process. Neural analysis indicated that high-frequency (12–100 Hz) power in Hp represented the temporal context states. Moreover, dynamic correlation and decoding results provided further support for the high-frequency dependence of model-based valuations. In addition, we observed a significant increase in hippocampal neural similarity at the low-frequency band (1–12 Hz) for common temporal context states after learning. Overall, our findings suggest that pigeons use model-based inferences to learn multi-step tasks, and multiple LFP frequency bands collaboratively contribute to model-based learning. Specifically, the high-frequency (12–100 Hz) oscillations represent model-based valuations, while the low-frequency (1–12 Hz) neural similarity is influenced by the relationship between temporal context states. These results contribute to our understanding of the neural mechanisms underlying model-based learning and broaden the scope of hippocampal contributions to avian behavior.

## 1. Introduction

Decisions in the natural world are rarely made in isolation. Each selected action by an organism influences the future situations it encounters, and these situations, in turn, impact future outcomes [1]. Therefore, when confronted with an unfamiliar multi-step task, we must acquire, retrieve, and update representations of relationships among different situations [2,3]. Subsequently, we construct an internal model of the environment and use it to plan behaviors in multiple steps into the future. This approach of guiding organism behavior by representing the relationships between states through learning is referred to as model-based (MB) learning, which bases choice on cognitive representations of the world and causal inference on environmental–behavioral structure [4]. Previous research on MB learning has generated important insights into the planning abilities of both humans and other mammals [5,6,7]. Extensive studies have explored the neural mechanisms responsible for MB learning, with a widely accepted recognition of the pivotal role played by the hippocampus in this process [8,9,10].

The Hp has long been theorized to function as a cognitive map of physical space, supporting navigational decision-making [3,7,11]. Classic experiments demonstrate hippocampal involvement in navigation tasks [12,13] as well as the existence of ‘place cells’, which both encode the current location [14] and ‘sweep out’ potential future paths at multiple timescales [15,16]. These findings have led to computational models positing the Hp’s crucial role in model-based learning [17,18,19]. A recent review suggested that multi-step planning shares neural mechanisms with navigation, indicating the Hp’s possible involvement in connecting multiple action–outcome associations [1]. The extensive research on the Hp in humans and other mammals has uncovered its contributions to model-based learning and potential neural mechanisms from various perspectives. On the other hand, numerous parallels in hippocampal structures between mammals and birds have been identified, including the expression of specific receptor subtypes and subunits, cell types, connectivity, and their role in cognitive learning [20,21,22,23]. However, little research has focused on the role of the avian Hp in non-spatial tasks, particularly in multi-step tasks. It remains unclear whether the avian Hp can also contribute to model-based learning and what its potential neural mechanisms might be.

To uncover the role of the Hp in model-based valuations and relational structural knowledge representations in pigeons during multi-step learning, we trained pigeons to perform a two-step task. The behavior performance and hippocampal multi-channel local field potentials (LFPs) were recorded during the task. We combined methods of the reinforcement learning model and neural signal analysis to explore the contribution of the pigeon hippocampus to model-based learning. Our research contributes to a better understanding of the neural mechanisms involved in avian multi-step decision-making.

## 2. Materials and Methods

### 2.1. Experimental Procedure and Data Acquisition

#### 2.1.1. Behavioral Task and Apparatus

Pigeons performed the two-step task in a custom-made bird-specific apparatus (Figure 1A). The apparatus, measuring 60 cm on all edges, was made of opaque acrylic material to prevent the animals from viewing the external environment. It was enclosed on five sides, with the remaining side functioning as an openable door. Positioned opposite the door was an 80 mm × 60 mm screen for presenting event cues, with three keys located below the cue presentation area. Additionally, there was a 30 mm high and 40 mm long opening below the screen through which the food reward was dispensed. To provide a stable standing surface for the pigeons, the bottom of the apparatus was covered with a stainless-steel mesh.

Six adult pigeons were trained to perform the two-step task. This task had been modified from the classic two-step decision-making task commonly used by humans and rats [1,24,25,26]. The schematic diagram of experimental tasks is shown in Figure 1B. This task consisted of two structures: the transition structure in the first step and the reward structure in the second step. After a 5 s intertrial interval (ITI), Step 1 was followed by the simultaneous presentation of two differently colored markers (S1^+^ and S1^−^) on both sides of the screen (the positions of the S1^+^ and S1^−^ markers were randomized to prevent the pigeons from having inertia preference for one side or the other). The pigeons were trained to choose one of the two options within 2 s and confirm their choice by pecking the key below the corresponding target option. A probabilistic transition took place, with the probability depending on the choice of the pigeons. Following this, a blue triangle (S2^+^) or a circle (S2^−^) marker appeared in the middle of the screen, indicating the outcome of the transition and initiating Step 2 (reward structure). In which an 80% chance of appearing S2^+^ and a 20% chance of appearing S2^−^ when S1^+^ was chosen, and a 20% chance of appearing S2^+^ and an 80% chance of appearing S2^+^ when S1- was chosen (the pre-set state transition probabilities were shown in Equation (1)). The pigeons were then trained to peck the key below the S2 stimulus within 2 s, followed by a reward delivered with specific probabilities—an 80% chance of receiving a 3 s reward when pecking the key corresponding to the S2^+^ and a 20% chance when pecking the key corresponding to the S2^−^ (the pre-set reward probabilities were shown in Equation (2)). The ‘+’ and ‘−’ in S^+^ and S^−^, respectively, indicated that the option or state was easier or more difficult to obtain rewards. Once the pigeon completed two stages of pecking keys, the trial was considered valid, regardless of whether they received a reward. Each trial was separated by an ITI marked by a gray screen. If the pigeon did not peck the key within 2 s of the stimulus presentation, no reward was delivered, and this was followed by the ITI period. A session comprised 50 trials, with a 10 min interval between two sessions. The pigeons were trained to complete 2 sessions each day, and each pigeon was ultimately trained to complete 60 sessions, with weekends off.
(1)$$P_{ss^{\prime }}=\left[\begin{array}{cc} P\left({S2}^{+}|{S1}^{+}\right) & P\left({S2}^{-}|{S1}^{+}\right)\\ P\left({S2}^{+}|{S1}^{-}\right) & P\left({S2}^{-}|{S1}^{-}\right) \end{array}\right]=\left[\begin{array}{cc} 0.8 & 0.2\\ 0.2 & 0.8 \end{array}\right]$$
(2)$$P_{r}=\left[\begin{array}{c} P\left(R|{S2}^{+}\right)\\ P\left(R|{S2}^{-}\right) \end{array}\right]=\left[\begin{array}{c} 0.8\\ 0.2 \end{array}\right]$$

#### 2.1.2. Subjects and Surgery

Six adult pigeons (450–550 g) of unknown sex were utilized as subjects and housed in a 3 m × 3 m × 2 m animal facility with ample sunlight and good ventilation. During the experiment, food was restricted, while drinking water was freely available. All animal-related experimental procedures were approved by the Life Science Ethical Review Committee of Zhengzhou University, with efforts taken to minimize the number of animals used and their suffering.

During the microelectrode array implantation surgical procedures, the animals were anesthetized with 3% pentobarbital sodium (0.14 mL/100 g body weight). The head was positioned in a customized stereotaxic holder for pigeons, with the anterior fixation point (i.e., beak bar position) set 45° below the horizontal axis of the instrument. Neural data were captured using a 16-channel tungsten microwire array (Figure 1C; Hongkong Plexon Inc., Hongkong, China), which was chronically implanted in the Hp area (coordinates: A 5.50 mm; L ± 1.00 mm; D 1.00–1.20 mm), as per the atlas provided by Karten and Hodos [27], Güntürkün, Onur [28].

#### 2.1.3. Acquisition of Neurophysiological Data

The pigeons recovered for one week after the surgery, and then the signals were recorded at a 2000 Hz sampling rate using a Multi-channel Acquisition Processor (Cerebus^TM^, 128 channels, Blackrock Microsystems, Salt Lake City, UT, USA) to simultaneously record 16-channel LFP signals from the Hp area. The LFPs were filtered using a 0–250 Hz Butterworth lowpass filter. Additionally, the behavior performance of the pigeons was monitored and recorded throughout the experiment.

#### 2.1.4. Histological Reconstructions

Following the completion of electrophysiological recordings, the recording sites were histologically verified in six animals using multiple electrolytic lesion marks in the striatum. Subsequently, the deeply anesthetized pigeons underwent microcurrent stimulation to damage the recording sites (current intensity: 1.2 mA, duration: 30 s, and a total of 3 repetitions). The electrically damaged pigeons were then perfused with 4% paraformaldehyde, and the brains were frozen and sectioned into 40 μm coronal slices. Hematoxylin and eosin (HE) staining was conducted on the brain sections. Finally, the actual implantation (damaged) sites were compared with the brain atlas locations to verify their validity (Figure 1C).

### 2.2. Dynamic Reinforcement Learning Model for Pigeon Behaviors

Model-based learning is a planning strategy that maintains separate estimates of the probability with which each choice (selecting the S1^+^ or the S1^+^ option) will lead to each outcome (the S2^+^ or the S2^−^ reward state becoming available), $T_{s1,s2}$, as well as the probability, $R_{s2}$, with which each outcome will lead to a reward. This strategy assigns values to the options by combining these probabilities to compute the expected probability with which the selection of each option will ultimately lead to a reward. We built a reinforcement learning model (RL model) based on successor representation (SR) learning [29,30,31,32] to simulate behavioral choices and estimate the evolution of option and state values acquired by pigeons. This model utilizes knowledge of transition and reward functions to compute value. The RL model comprises two main components: (1) estimating the probability of state transition and (2) estimating the probability of reward.

The RL model uses two structures to compute state values: a reward vector $R_{s^{\prime }}$ and a transition matrix $T_{s,s\prime }$. $R_{s^{\prime }}$ stores the immediate reward expected upon encountering state s′. $T_{s,s\prime }$ stores the expected frequency of (future discounted) visits to the state s′ along a sequence of states starting in state s.
(3)$$T_{s,s^{\prime }}=E\left[\sum_{t=0}^{\infty }\;\gamma^{t}I\left(s_{t}=s^{\prime }\right)|s_{0}=s\right]$$
(4)$$R_{s^{\prime }}=E\left[\gamma I\left(r=1\right)|s_{0}=s^{\prime }\right]$$
where E denotes expectation, I(·)=1 if its argument is true, and 0 otherwise, $s_{t}$ is the state encountered t time-steps following the visit of state $s_{0}$ and γ is a discount or forgetting parameter. Since our task has only one step from the initial state s to the final state s’, we set t = 1. The RL model combines these two structures to compute the value of a state s, $Q_{s}$, in trial i, by taking the inner product of $R_{s^{\prime }}$ and the row of $T_{s,s^{\prime }}$ corresponding to that state:(5)$$Q_{S}\left(i\right)=\sum_{s^{\prime }}\;T_{s,s^{\prime }}\left(i\right)\times R_{s^{\prime }}\left(i\right)$$
(6)$$Q_{s^{\prime }}\left(i\right)=R_{s^{\prime }}\left(i\right)$$

For the two-step task in this study, the option value of the first step was denoted as $Q_{S1}$, in which the option value of S1^+^ is denoted as $Q_{{S1}^{+}}$, and the option value of S1^−^ is denoted as $Q_{{S1}^{-}}$, the state value corresponding to Step 2 is denoted as $Q_{S2}$, which also includes $Q_{{S2}^{+}}$ and $Q_{{S2}^{-}}$. The calculation method is as follows:(7)$$Q_{S1}\left(i\right)=\sum_{S2}\;T_{S1,S2}\left(i\right)\times R_{S2}\left(i\right)$$
(8)$$Q_{S2}\left(i\right)=R_{S2}\left(i\right)$$

The probability that the animal would choose the option j in trial i, $P_{j}\left(i\right)$, was determined from option value functions according to the softmax transformation as Equation (9):(9)$$P_{j}\left(i\right)=\frac{\exp\left(\beta Q_{j}\left(i\right)\right)}{\sum_{j=1}^{K}\exp\left(\beta Q_{j}\left(i\right)\right)}$$
where β is the ‘inverse temperature’ parameter that controls the level of stochasticity in the choice, ranging from β=0 for completely random responding and β=∞ for deterministically choosing the highest value option.

Considering the longer duration of the two-step learning task for pigeons, we compared the performance of dynamic and static models in simulating the actual behavior of pigeons. The dynamic model involves splitting the entire learning process into N windows and fitting parameters β(n) and γ(n) within each window n, while the static model maintains a fixed learning process $\theta_{m}$.

We performed maximum a posteriori fits using weakly informative priors on all parameters [24,33]. The prior parameters $\beta_{0}$ was a Gamma distribution with a = 1.2, b = 5.0, i.e., ${\theta (\beta }_{0})\sim Gamma(1.2,5.0)$, and prior parameters $\gamma_{0}$ was a Beta distribution with a = b = 1.1, i.e., ${\theta (\lambda }_{0})\sim Beta(1.1,1.1)$. The RL model was fit to the behavioral data of each animal separately. This produced a set of model parameters (i.e., forgetting parameter γ and inverse ‘temperature’ β) for each animal.

### 2.3. LFP Analysis

#### 2.3.1. Preprocessing of LFP Data

The LFP signals were preprocessed in the following three steps: (1) The Sample rate of LFPs was reduced to 1 KHz using the median down-sampling algorithm. (2) Correlated noise between electrodes was removed by using the adaptive common average reference method [34]. (3) The adaptive filter based on the LMS algorithm was used to remove 50 Hz power frequency interference [35]. In addition, for those trials with a reaction time of less than 0.5 s and wings flapping greatly, they were removed. Overall, the final dataset consisted of 480 sessions for 6 pigeons (60 sessions for one pigeon). Then, in the subsequent data analysis, LFP time series were epoched and aligned on action in Step 1’s onset. Baseline activity’s LFP data were taken in a period from −2.5 s to −2.0 s relative to action choice Step 1 onset, the choice period from −1 s to 0 s, the S2 period from 0 s–1 s, which ensures that the pigeon has noticed the transition state.

#### 2.3.2. Time-Frequency Power Estimates of LFPs

To investigate the hippocampal neural encoding patterns of two temporal context states during the pigeons’ two-step task learning, we used the Morlet wavelet [36] method to estimate the time-frequency power of two states across all trials. The power spectra were computed on 50 frequency steps, logarithmically spaced between 1 Hz and 100 Hz. For each trial, we computed the time-frequency power for each electrode separately within a period from 1 s before the presentation of S1 to 1 s after the presentation of S2 and saved the results as a power matrix (Figure 1D,E). The selected temporal window encompasses the pecking moments in Step 1, signifying the pigeon’s awareness of the two states, S1 and S2. It is important to note that we considered two potential sources of unreliable results: (1) motion artifacts caused by the pigeons’ pecking and (2) temporal variations in the baseline of the local field potentials (LFPs) during the learning process. To address these issues, we conducted the following operations in the subsequent analysis: firstly, we eliminated motion artifacts by excluding the data 50 ms before and after each pecking moment; secondly, we normalized the LFP time-frequency power relative to the baseline of all electrodes for each trial by dividing the power matrix of the selected period by the baseline matrix of the same period.

#### 2.3.3. Analysis of Neural Similarity Representations in Temporal Context States

To study how the neural similarity for two temporal context states in the hippocampal of pigeons changes with the increase in behavioral accuracy during learning, we quantified the similarity of neural representations during S1 presentation (period of 1 s before key pecking in Step 1) and S2 presentation (period of 1 s before key pecking in Step 2) by comparing epochs of brain activity at Hp electrodes via representational similarity analysis [37,38,39].We binned continuous time wavelet transforms into 100 ms epochs spaced every 20 ms (80% overlap) and averaged the instantaneous time-frequency power over each epoch. To account for changes in power across experimental trials, we z-transformed power values separately for each frequency and each trial using the mean and SD of all 100 ms epochs for that trial. For each temporal epoch, we subsequently averaged the z-transformed power across six frequency bands: Delta (1–4 Hz), Theta (4–8 Hz), Alpha (8–12 Hz), Beta (12–30 Hz), Low gamma (30–60 Hz), and High gamma (60–100 Hz).

For every temporal epoch in each trial, we constructed a feature vector composed of the average z-scored power for every electrode and every frequency band (Figure 1E). For each S1 presentation temporal epoch, i, and for each S2 presentation temporal epoch, j, we define feature vectors as follows:(10)$$\begin{array}{c} {\vec{S1}}_{i}|=\left[z_{1,1}\left(i\right)\ldots z_{1,F}\left(i\right)\ldots z_{L,F}\left(i\right)\right]\\ {\vec{S2}}_{j}|=\left[z_{1,1}\left(j\right)\ldots z_{1,F}\left(j\right)\ldots z_{L,F}\left(j\right)\right] \end{array}$$
where $z_{l,f}(i)$ is the *z*-transformed power of electrode l=1 . . . L at frequency band f=1 . . . F in temporal epoch i. For L electrodes and F frequency bands, we thus create a feature vector at each temporal epoch that contains K=L∗F features.

To quantify the neural similarity of temporal context states during trial n, we calculated the cosine similarity between all state S1 and state S2 feature vectors ${\vec{S1}}_{i}$ and ${\vec{S2}}_{j}$ for all pairs of S1 presentation and S2 presentation temporal epochs during that trial (Figure 1F). Cosine similarity gives a measure of how close the angles of two vectors are in a multidimensional space. For each trial, n, we generate a temporal map of similarity values rho-ρ (Figure 1G):(11)$$\rho_{n}\left(i,j\right)=\frac{{\vec{S1}}_{i}\cdot {\vec{S2}}_{j}}{\left\|{\vec{S1}}_{i}\right\|\left\|{\vec{S2}}_{j}\right\|}$$
where $\rho_{n}\left(i,j\right)$ corresponds to the neural similarity of temporal context states across all electrodes and all frequencies between S1 epoch i and S2 epoch j during trial n. For every pigeon, we compute the neural similarity maps separately for all common states (S1^+^ to S2^+^) and uncommon states (S1^+^ to S2^−^).

### 2.4. Correlation between Hippocampal LFPs Power and Model-Based Valuation

#### 2.4.1. Regression Analysis

We used a one-variable linear regression model to investigate the relationship between the hippocampal neural activity of pigeons and model-based valuation, specifically the option values in Step 1 and state values in Step 2 estimated by the RL model. Our analysis focused on the correlation between model-based valuation and normalized hippocampal LFPs power during the pigeons’ learning process. The LFPs power served as the dependent variable ($y_{S1}\;or\;y_{S2}$) calculated as the average normalized power within the 0.5 s period before pecking. The option value, $Q_{S1}$, and state value, $Q_{S2}$, were estimated by the RL model and used as the independent variables.

Formula (12) examines the relationship between changes in normalized LFP power in the Hp during S1^−^ and S1^+^ presentation and the estimated option values $Q_{{S1}^{+}}$ or $Q_{{S1}^{-}}$ during pigeons’ learning process. Formula (13) investigates the correlation between changes in normalized LFP power in the Hp during S2^−^ and S2^+^ presentation and the estimated state values $Q_{{S2}^{+}}$ or $Q_{{S2}^{-}}$ during pigeons’ learning process.
(12)$$\begin{array}{c} y_{{S1}^{+}}\left(t\right)=\beta_{0}+\beta_{1}Q_{{S1}^{+}}\left(t\right)\\ y_{{S1}^{-}}\left(t\right)=\beta_{0}+\beta_{1}Q_{{S1}^{-}}\left(t\right) \end{array}$$
(13)$$\begin{array}{c} y_{{S2}^{+}}\left(t\right)=\beta_{0}+\beta_{1}Q_{{S2}^{+}}\left(t\right)\\ y_{{S2}^{-}}\left(t\right)=\beta_{0}+\beta_{1}Q_{{S2}^{-}}\left(t\right) \end{array}$$
where, $y_{{S1}^{+}}\left(t\right)$ and $y_{{S1}^{-}}\left(t\right)$ are the mean normalized LFP power in a period from 0.5 s before S1^+^ action choice or S1^−^ action choice in session $t,\;y_{{S2}^{+}}\left(t\right)$ and $y_{{S2}^{-}}\left(t\right)$ are the mean normalized power in a period from 0.5 s before S2^+^ pecking conformation or S2^−^ pecking conformation in session t, $Q_{{S1}^{+}}\left(t\right)$ and $Q_{{S1}^{-}}\left(t\right)$ are respectively the mean option value estimated by the RL model in session t, $Q_{{S2}^{+}}\left(t\right)$ and $Q_{{S2}^{-}}\left(t\right)$ are respectively the mean state value estimated by the RL model in session t. $\beta_{0-1}$ are the regression parameters.

#### 2.4.2. Behavior Choice Decoding

To further demonstrate that the hippocampal neural activity pattern of pigeons contributes to the model-based valuation, we used a Support Vector Machine (SVM) [40] to decode the pigeon’s behavioral choice in Step 1. The input for the decoding model encompassed the normalized power of all S1^+^ and S1^−^ trials for each session, with 70% allocated to the training set and 30% to the test set. The decoding was executed through 10-fold cross-validation. We repeated the decoding analyses until all data were utilized at least once, and the decoding accuracies from multiple iterations were averaged for each round.

### 2.5. Statistical Analysis

For the regression analysis of the study, statistical significance analysis on the regression equation was conducted using a one-way ANOVA, and the regression coefficients were evaluated through paired *t*-tests. The Normality of distributions for each dataset was assessed using the Lilliefors test to determine whether to use parametric or non-parametric tests for the statistical analysis of other study results. Paired *t*-tests were utilized for data conforming to normal distribution, while the Wilcoxon rank-sum test was employed for data not conforming to normal distribution. The statistical results are predominantly presented as mean ± standard deviation (SD) or mean ± standard error (SE), as specified in the figure legend, with a significance level set at 5%. Values were considered to indicate significant differences at *p* < 0.05. The statistical analyses were performed using Matlab 2022a, GraphPad Prism 9.5.0, and OriginPro 2023 learning edition. In the result figures, “*” means *p* < 0.05, “**” means *p* < 0.01, and “***” means *p* < 0.001.

## 3. Results

To examine the potential role of the pigeon’s Hp in model-based valuations and temporal context association, six pigeons were trained to complete a two-step learning task (Figure 1B). In the first step of the task, known as the transition structure, the pigeons chose between two options (S1^+^ and S1^−^), distinguished by different colors, each of which leads to one of two reward states (S2^+^ or S2^−^), distinguished by different shapes, becoming available with a probability of 80% (common transition) and to the other reward state becoming available with a probability of 20% (uncommon transition). In the second step, referred to as the reward structure, the pigeons had no choice but to peck the key below the S2, resulting in a reward with an appropriate probability. These transition probabilities constituted stable relationships between actions (choice) and their likely outcomes (reward). Subjects, therefore, had the opportunity to incorporate these action–outcome relationships into an internal model and to use them to plan. We recorded 16-electrode local field potentials (LFPs) from the hippocampi of six pigeons (Figure 1D) during the completion of the two-step task, with the recording sites histologically confirmed in all animals (Figure 1C).

### 3.1. Behavioral Results

The evolution of behavioral performances shows that pigeons started to choose the option with a higher reward probability (S1^+^) more frequently and obtained more rewards after 10 days and 20 sessions (Figure 2A,B). This suggests that they had learned the relative future reward expectations after discounts between options S1^+^ and S1^−^. Next, we fitted two parameters, γ and β, of the dynamic RL model and showed their changes in Figure 2C. Our findings indicate a progressive increase in the values of γ and β in the dynamic RL model over the course of learning. As learning progresses, the γ value eventually approaches one, signifying that the pigeon gradually establishes relationships between states. The increasing β value also indicates that the pigeon becomes more confident in the estimated values from the model, gradually transitioning from a random choice (exploration) to a value-based choice (exploitation). Figure 2D demonstrates that the dynamic model-based learning approach better captures the animals’ behavioral strategy during the learning process. Finally, using the dynamic model-based learner, we estimated the subjective option values ($Q_{{S1}^{+},\;}Q_{{S1}^{-}\;}$) in Step 1 and state values ($Q_{{S2}^{+},\;}Q_{{S2}^{-}\;}$) in Step 2 of the pigeon’s session-by-session (Figure 2E,F). It was observed that the option values in Step 1 and the state values in Step 1 learned by the model-based approach gradually converged towards our pre-set reward expectations. Specifically, the correct (S1^+^) choice rate exceeded 90% after approximately 30 sessions, while the estimated option value of S1^+^ reached 0.68 (the pre-set reward expectation), and the state value of S2^+^ reached 0.8 in the same session. The findings indicate that the pigeons successfully acquired the entire task. In conclusion, this behavioral modeling offers two crucial insights into our study. Firstly, the dynamic model-based learners adeptly simulate the actual learning behavior of pigeons. Secondly, the RL model provides an approximate estimation of pigeons’ subjective option values in Step 1 and state values in Step 2, trial by trial.

### 3.2. The Temporal Context States Modulate the Hippocampal LFPs High-Frequency Power

We then investigated whether modulations in hippocampal LFP activity differed among high-reward and low-reward probability states in two-step task learning. We calculated hippocampal LFPs normalized power distribution of all electrodes in different frequency bands before key pecking in Step 1 and Step 2, respectively. We considered the differences in the results between common transition with high reward probability (S1^+^ to S2^+^) and low reward probability (S1^−^ to S2^−^) conditions. We observed a significant increase in the normalized power distribution of Beta (12–30 Hz) and Gamma (30–80 Hz) bands in the hippocampus during the late learning stage compared to the early stage under the conditions of state S1^+^ and S2^+^ (Figure 3A). However, the normalized power of S1^−^ and S2^−^ with lower reward probability did not show a synchronous increase with learning. Specifically, the power increase only occurred under the S2^−^ condition in Step 2 (Figure 3B). Conversely, the lower bands (1–12 Hz) did not exhibit the same phenomenon for all conditions; only a few electrodes showed significant differences (Figure 3C,D). Additionally, an analysis of the dynamic changes in normalized power during the entire learning process (Figure 3E,F) revealed a gradual, synchronous increase in normalized power for S1^+^ and S2^+^ over the learning process. Notably, the normalized power of the state S2^+^ close to the reward was consistently higher than that away from the reward (S1^+^) (Figure 3E). These phenomena were not observed at low frequencies (Figure 3G,H). These findings indicate that the hippocampal high-frequency power of pigeons is influenced by temporal context-dependent states that elicit more rewards in multi-step tasks.

### 3.3. High-Frequency Power in the Hp Correlates of Model-Based Valuations

One of the main goals of the study was to investigate the relationship between Hp neural activity and model-based valuations in a two-step task. Therefore, we next fitted a univariate linear regression model session-by-session to describe the dynamic relationship between hippocampal LFPs high-frequency band power under different states and the model-based valuations ($Q_{{S1}^{+}},\;Q_{{S1}^{-}}$, $Q_{{S2}^{+}},\;Q_{{S2}^{-}}$). Table 1 and Table 2 show the results of the statistical analyses. We observed a robust, time-sensitive linear correlation between the normalized high-frequency band power of S1^+^ in Hp and the option value $Q_{{S1}^{+}}$ in Step 1 (Figure 4A), but this correlation was not observed at the 1–12 Hz band (Table 1, left). Conversely, there was no significant correlation between the normalized power of S1^−^ in Hp and option value $Q_{{S1}^{-}}$ in Step 1 in either frequency band (Figure 4C; Table 1, right). Furthermore, a strong, time-sensitive linear correlation was observed between the normalized power of S2^+^ in Hp and the state value $Q_{{S2}^{+}}$ in Step 2 at any frequency band (Figure 4B; Table 2, left). Similarly, the normalized LFPs power of S2^−^ in Hp exhibited a significant correlation with the state value $Q_{{S2}^{-}}$ in the high-frequency band (Figure 4D; Table 2, right). Further, we observed that the regression results of Step 2 were better than those of Step 1 in almost all conditions. These results further suggest that the high-frequency power was modulated by the value of states or options that may bring more rewards.

Next, we attempted to decode the action choice of pigeons in Step 1 using Hp normalized power as features. The choice decoding results of six pigeons in different frequency bands are shown in Table 3. As expected, the high-frequency power distribution in the Hp can well decode pigeons’ behavior choices in Step 1, with the average accuracy of all pigeons’ behavioral choices reaching 81.4%. However, the decoding accuracy of low-frequency (1–12 Hz) power was not good. These results indicate that the representation of valuations estimated by model-based learning in the Hp of pigeons is highly dependent on high frequencies.

### 3.4. Low-Frequency Dependence of the Relationship between Temporal Context States

Model-based decisions rely on an organism’s ability to establish a relationship between discrete states. In this study, we investigated the changes in neural similarity (NS) in the Hp as pigeons acquired knowledge of state transitions. We examined both common (S1^+^ to S2^+^) and uncommon (S1^+^ to S2^−^) conditions during the time window of S1 and S2 presentations in the early and late learning stages across different frequency bands. We observed an increase in neural similarity in common transition trials at the low-frequency band (1–12 Hz) after learning but no significant increase in neural similarity in uncommon transition trials (Figure 5A,B; analysis was locked to the key pecking in Step 1). The difference analysis between common and uncommon transition trials indicated that the increase in neural similarity occurred during the time windows of S1^+^ presentation (~−0.6–0 s) and S2^+^ presentation (~0.2–0.8 s) (Figure 5C–E,H, t(200) = 4.01, *p* < 0.0001). This time, the region of interest is defined as tROI 1. In addition, the difference analysis between early and late trials showed that the increase in neural similarity only occurred in common transition trials (Figure 5F,G,I; t(200) = 5.15, *p* < 0.0001). This time region of common transition trials is defined as tROI 2. These two time regions are close to the moment when the pigeon makes a choice in Step 1 and confirms the transition state in Step 2, and the increase in neural similarity in these time regions may indicate that the pigeon has predicted the next transition state when making a choice in Step 1. In contrast, we found no evidence of an increase in neural similarity for uncommon transition trials (S1^+^ to S2^−^) in the Hp, both in the early and late stages of learning (Figure 5G; all time windows or clusters, *p* > 0.05), as well as for common transition trials in the early learning stage (Figure 5D; all time windows or clusters, *p* > 0.05).

Moreover, we observed that the removal of 1–12 Hz activity significantly decreased neural similarity in the common/uncommon cluster (corresponds to tROI 1 and tROI 2 in Figure 5, under all conditions, t < 1; *p* > 0.5, Figure 6A–I). Overall, these findings suggest that the increased neural similarity in Hp is regulated by the temporal context state transition knowledge, and the representation of temporal context states relationship in Hp exhibits low-frequency dependence.

## 4. Discussion

The study aimed to investigate the contribution of the avian Hp to model-based learning and its potential neural mechanisms. For this purpose, we recorded the 16-channel LFP signals from the Hp while the pigeons performed a two-step learning task. We examined the correlation between the power distribution in the Hp and model-based valuations. The neural analysis and decoding results supported the high-frequency (12–100 Hz) dependence of model-based valuations. Furthermore, our findings indicate a significant increase in hippocampal neural similarity in the low-frequency band (1–12 Hz) under the common temporal context states condition after learning. These findings indicate that both low and high-frequency power distributions in the pigeon’s Hp work together to facilitate model-based learning, including valuation and structural relationship building.

We reported the successful adaptation of the two-step task—a repeated-trial, two-step decision task widely used in human and rat research—for pigeons. Analysis of behavior data on our task reveals a dominant role for model-based learning of pigeons, which is consistent with earlier results obtained in humans and rats [10,41]. Notably, our analysis reveals that pigeons take a long time to learn optimal action (each pigeon needs an average of 400 tries to learn the optimal action), in contrast with the performance of human subjects or rats on the same task [10,24], which in turn allows us to analyze the dynamic relationship between neural activity in Hp and behavioral computation of the complete learning process in pigeons with no change in the task. We observed that a model with static parameters could not capture the behavior of pigeons, but the model with time-varying parameters can capture the behavior of pigeons well (Figure 2C); that is, the value of discount rate γ gradually increases and finally approaches one (Figure 2D). In computational theory, γ is used to regulate the effect of immediate or future rewards on the current actions. The larger γ is, the more the agent focuses on long-term future gains. Therefore, when the agent has fully established the model of the environment, γ equals one. Recent developments in computational theory have led to claims that time-varying γ outperforms fixed gamma in terms of performance [42]. Overall, despite using a simple two-step task, we still demonstrated that pigeons use model-based learning rules.

Planning can be defined as action selection that leverages an internal model of the outcomes likely to follow each possible action. In the present study, we noticed an increase in high-frequency (12–100 Hz) power in the Hp accompanying the improved accuracy of model-based valuations (Figure 3 and Figure 4), which indicates that the Hp of pigeons contributes to valuations in model-based learning. This aligns with prior research on rats and humans, emphasizing the significant role of this region in model-based planning [9,41,43,44,45]. Additionally, we found that the high-frequency power distribution in the Hp can predict the behavioral choices of pigeons in the two-step task (Figure 4E), underscoring the significance of high-frequency power distribution as an indicator of model-based valuation in pigeons. It is notable that while previous studies confirmed the role of gamma in reward-driven learning in mammals [46,47], no reports have indicated enhanced high-frequency power in the Hp of mammals contributing to multi-step planning. These studies indicate that the Hp of mammals and avian plays an important role in model-based learning, but their underlying neural mechanisms show species specificity.

Structural relationship knowledge is crucial for model-based learning, which requires the brain to remember the associations of discrete events and store this knowledge. Here, we indicated that low-frequency (1–12 Hz) neural similarity in the Hp increased with the establishment of structural relationships in model-based learning. Specifically, we observed an increase in neural similarity during common state transitions after learning, as opposed to uncommon state transitions (Figure 6). This observation aligns with the phenomenon identified in prior spatial and non-spatial task learning [38,39,48]. Hippocampal neurons have been observed to encode sequences of spatial locations under various experimental conditions [17,49]. Additionally, earlier studies have indicated an increase in neural similarity of distributed cortical oscillations at low-frequency bands during successful memory retrieval in both temporal and spatial contexts [38,39]. Furthermore, the Hp was confirmed to represent the sequential relationships among an extended sequence of non-spatial events [50]. The above research results demonstrate the consistency of the contributions of avian and mammalian Hp to the establishment of structural relationships in model-based learning.

In summary, we observed the hippocampal neural patterns of the pigeon in model-based learning based on the power distribution of different frequency bands from multiple electrodes. The results indicate that the low and high-frequency power in the pigeon Hp co-encode model-based learning. However, this study still has the following limitations. Firstly, we have not yet subdivided the pigeon hippocampal region to understand their specific contributions to model-based learning. Current research on mammals generally indicates that the dorsal Hp plays a dominant role in model-based learning [10]. Whether pigeons yield similar results will be subject to continuous investigation in our subsequent studies. In addition, in the current study, we only focused on the contribution of the pigeon Hp to the knowledge of structural relationships in tasks with fixed reward probabilities without exploring its role in adapting to new environments during task transitions. Future research will center on the hippocampus’s representation of relational knowledge under varied environmental conditions. Finally, biological learning of complex tasks usually cannot solely rely on a single brain region [51]; it often requires the collaborative efforts of multiple brain regions. It is widely believed that the Hp and the striatum collaboratively participate in learning a series of complex tasks, with the striatum primarily supporting model-free learning and the Hp primarily supporting model-based learning [6,24,45]. In future research, we will explore the role of the striatum in two-step tasks and investigate the collaborative effects of these two brain regions. In addition, our research results support the involvement of the Hp in model-based valuations. This is confirmed by the strong correlation between neural activity in pigeons during the choice period and model-based valuations, as well as the ability to decode pigeon selection based on neural activity. However, the current research lacks the quantification of the planning index. Therefore, in future studies, we will integrate the planning index to investigate the contribution of the pigeon Hp to model-based learning and its potential neural mechanisms.

Furthermore, in future research endeavors, we will aim to extend the current understanding by integrating more sophisticated and dynamic multi-step decision-making tasks. This could involve scenarios with variable reward probabilities, introducing unpredictability and complexity that more closely mimic real-life situations pigeons may encounter. The integration of such variables would not only test the adaptability and robustness of the pigeon’s Hp in changing environments but also offer deeper insights into the cognitive and neural processes underpinning decision-making in birds. Moreover, the establishment of behavioral models based on the insights gleaned from this study could enhance our understanding of animal behavior in general. These models could serve as frameworks for interpreting a wide range of animal actions, particularly in how they plan, make decisions, and learn from their environment.

## 5. Conclusions

This study aimed to investigate the role of the avian hippocampus (Hp) in model-based learning and its associated neural mechanisms. Neural analyzing in the Hp of pigeons during a two-step learning task revealed that both low (1–12 Hz) and high-frequency (12–100 Hz) power distributions contribute to model-based decision-making, including structural relationship building and valuation. Specifically, behavioral analysis showed that a model-based learner with time-varying parameters can capture well pigeon behavior, indicating a gradual increase in the value of the discount rate. Additionally, neural analysis indicated that high-frequency power in the Hp accompanied improved accuracy of model-based valuations, thereby contributing to model-based decisions. Furthermore, an increase in low-frequency neural similarity between temporal context states in the Hp was observed concomitant with the establishment of structural relationships. Our findings highlight the significant role of the avian Hp in model-based learning and its distinctive neural mechanisms.

## Figures and Tables

**Figure 1 animals-14-00431-f001:**
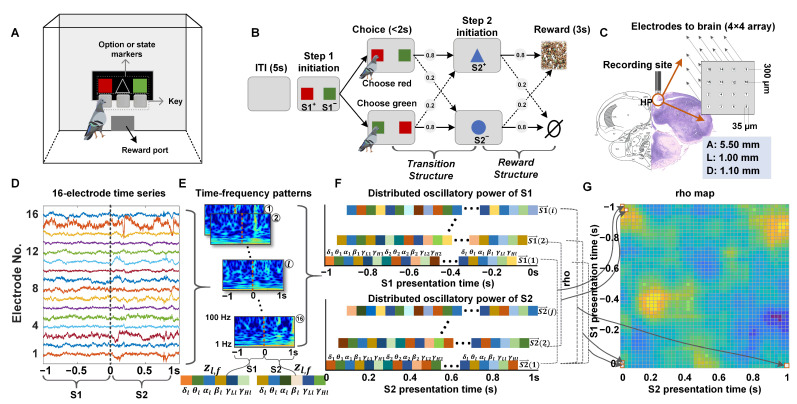
The apparatus, two-step probabilistic learning task, recording sites, and LFPs analysis. (**A**) Schematic diagram of the pigeon training apparatus. (**B**) The two-step task for pigeons is as follows: A 5 s gray screen illuminates to indicate the trial is ready. Step 1 (transition structure) is initiated by the simultaneous presentation of two differently colored markers (S1^+^ and S1^−^) on both sides of the screen. The pigeons indicate their choice by pecking the key below the corresponding target option. A probabilistic transition takes place, with probability depending on the choice of the pigeons. Following this, a blue triangle (S2^+^) or a circle (S2^−^) marker appears in the middle of the screen, indicating the outcome of the transition and initiating Step 2 (reward structure). The pigeons then peck the key below the S2 target within 2 s. A 3 s reward is delivered with an appropriate probability. (**C**) The histological verification of the implantation site. (**D**) The schematic of the acquired 16-electrode LFP signals, represented in different colors. (**E**) Time-frequency patterns (1–100 Hz) were extracted from all electrodes. For every temporal epoch during the S1 and S2 presentation, the z-scored power from every electrode l and every frequency f band was combined to create a single feature vector $Z_{l,f}$ for that epoch. Numbers 1–16 represent the electrode’s ID (**F**) Distributed oscillatory power of S1 (top panel) or S2 (bottom panel). Two feature vectors ${\vec{S1}}_{i}$ and ${\vec{S2}}_{j}$ composed of all vectors $Z_{l,f}$, for each S1 presentation temporal epoch i, and each S2 presentation temporal epoch j, were constructed. (**G**) Neural similarity (cosine similarity rho) for all temporal context-dependent state pairs was shown for a single trial.

**Figure 2 animals-14-00431-f002:**
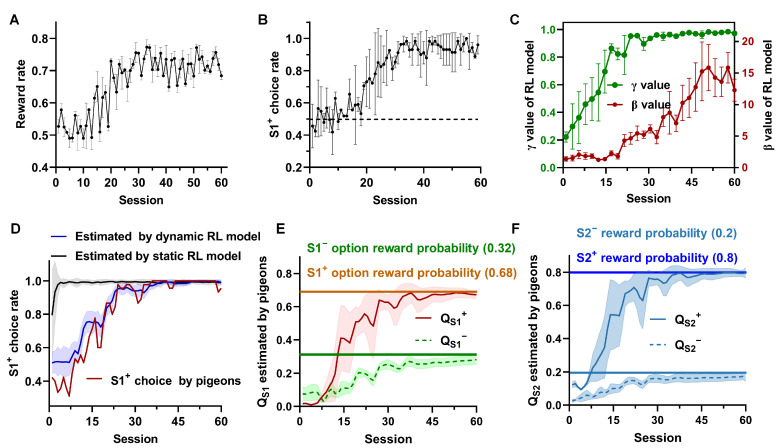
Pigeon’s behavior performance, parameter changes in the fitted RL model, option values, and state values estimated by dynamic RL model. (**A**) Reward rate of all pigeons during the whole learning process, where the reward rate of each session is calculated as the ratio of reward trials to the total number of trials (n = 6, session = 60, mean ± SD). (**B**) Dynamic correct choice (S1^+^) rate of all pigeons, with the value representing the ratio of S1^+^ choice trials to the total number of trials in each session (n = 6, session = 60, mean ± SD). (**C**) The changing trend of the parameters (γ, β) of the RL model fitted from all the pigeons’ behavior (mean ± SD). (**D**) The S1^+^ choice rate computed by one example pigeon (P090), estimated by dynamic RL model, and estimated by static RL model (The model ran 50 rounds, mean ± SD). (**E**,**F**) are the trends of the option value $Q_{S1}$ in Step 1 and state value $Q_{S2}$ in Step 2 estimated by RL model (n = 6, mean± SD).

**Figure 3 animals-14-00431-f003:**
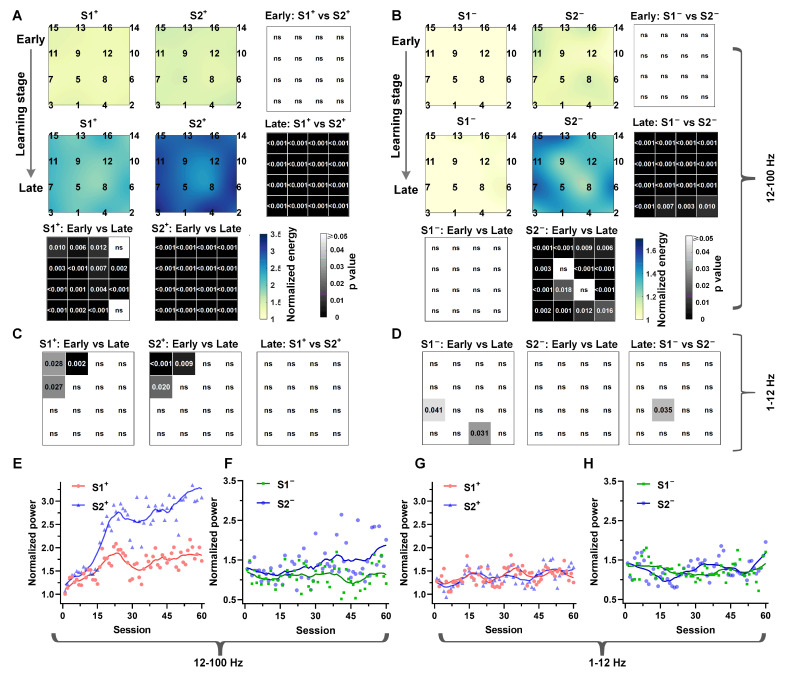
Hippocampal LFPs power distribution of temporal context-dependent states. Panels (**A**,**B**) display the normalized power distribution maps within the 12–100 Hz range for the common and uncommon temporal context-dependent states, respectively. Numbers 1–16 in the color map represent the electrode’s ID, and their arrangement corresponds to the relative position of the electrode implantation in the brain. The grayscale map corresponds to the significant differences of all electrodes under different conditions (n = 6, Early stage = 100 trials, Prob (S1^+^ choice) < 65%; Late stage = 100 trials, Prob (S1^+^ choice) > 90%). Panels (**C**,**D**) present the p-map for all different conditions in the low-frequency band (1–12 Hz), where “ns” indicates no significance. Panels (**E**,**F**) depict the dynamic changes in the normalized power in the hippocampus within the 12–100 Hz band, while panels (**G**,**H**) detail the dynamic changes within the 1–12 Hz band for the common temporal context states. The solid line in panels (**E**–**H**) indicates the average of the normalized power in each session (n = 6, session = 60) of every state. The data were smoothed with a five-point moving average. The normalized power of each trial represents the average within a 0.5 s time window before pecking the key in Step 1 or Step 2, and the result of each session is the average of the normalized power of all trials within that session.

**Figure 4 animals-14-00431-f004:**
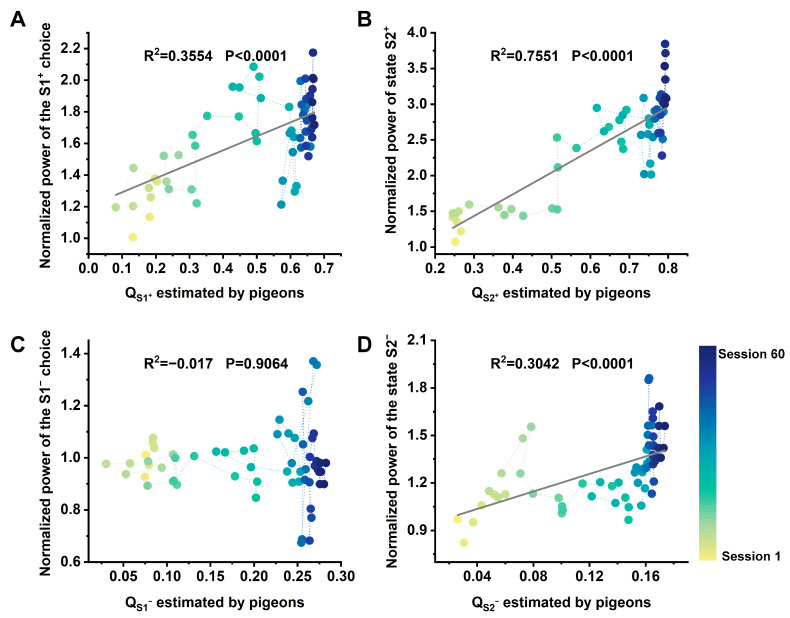
The hippocampal LFPs normalized high-frequency power of options in Step 1 and states in Step 2 are dynamically correlated with model-based valuations. (**A**) The relationship between the normalized power of option S1^+^ and the option value $Q_{{S1}^{+}}$ estimated by model-based. (**B**) The relationship between the normalized power of state S2^+^ and the state value $Q_{{S2}^{+}}$ estimated by model-based. (**C**) The relationship between the normalized power of option S1^−^ and the option value $Q_{{S1}^{-}}$. (**D**) The relationship between the normalized power of state S2^−^ and the state value $Q_{{S2}^{-}}$ estimated by model-based. The dots’ color fading from yellow to blue represents the passage from early sessions to late sessions.

**Figure 5 animals-14-00431-f005:**
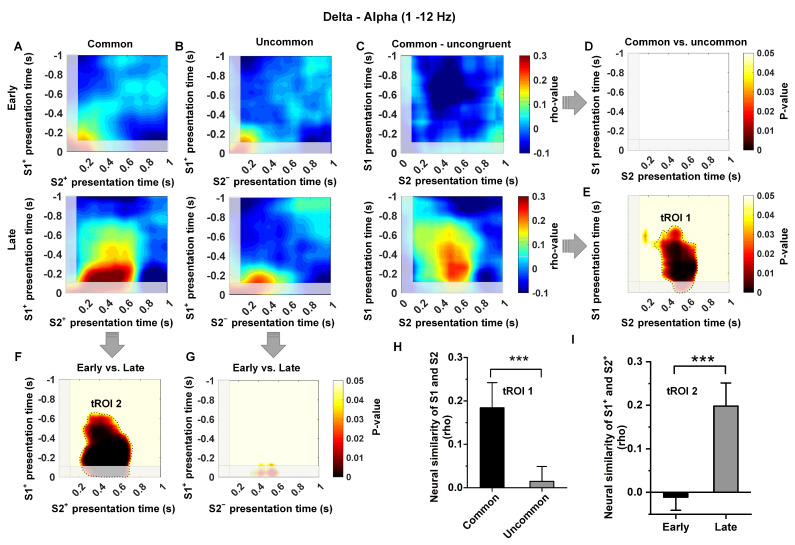
The neural similarity in the low-frequency band (1–12 Hz) in the Hp is influenced by common temporal context states. The grand average neural similarity map for common (Panel (**A**)) and uncommon (Panel (**B**)) trials in the Hp during the early and late learning stages. The time window aligns with the pecking key moment in Step 1, corresponding to 0 s in the panel. Panel (**C**) represents the difference between Panels (**A**,**B**). Panel (**D**) displays the p-map of the common vs. uncommon contrast in the early learning stage, with no significant clusters identified. In Panel (**E**), the *p*-map of the common vs. uncommon contrast reveals a significant cluster at *p*(corr) < 0.05 (outlined in black and named tROI 1) in the late learning stage, indicating that the neural similarity in the Hp represents the binding of temporal context states information (the relationship between two temporal context states). Panel (**F**) shows a P-map of the early vs. late learning stage contrast, demonstrating a significant cluster at *p*(corr) < 0.05 (outlined in black and named tROI 2) in the common states, indicating that the neural similarity in the Hp only represents the binding of common temporal context states. Panel (**G**) represents the early vs. late learning stage in the uncommon states, does not display any significant clusters. Gray shadow regions during the S1 presentation (−0.9–0 s) and during the S2 presentation (0–0.1 s) are not considered across the map due to the potential presence of pecking key artifacts. Panel (**H**) presents the neural similarity in both common and uncommon conditions in the late learning stage, in the tROI 1 of Panel (**E**). Panel (**I**) shows the neural similarity in both the early and late learning stages under the common condition in the tROI 2 of Panel (**F**). (n = 6; Early stage = 100 trials, Prob (S1^+^ choice) < 65%; Late stage = 100 trials, Prob (S1^+^ choice) > 90%; two sample test; *** indicates *p* < 0.001; mean ± sem).

**Figure 6 animals-14-00431-f006:**
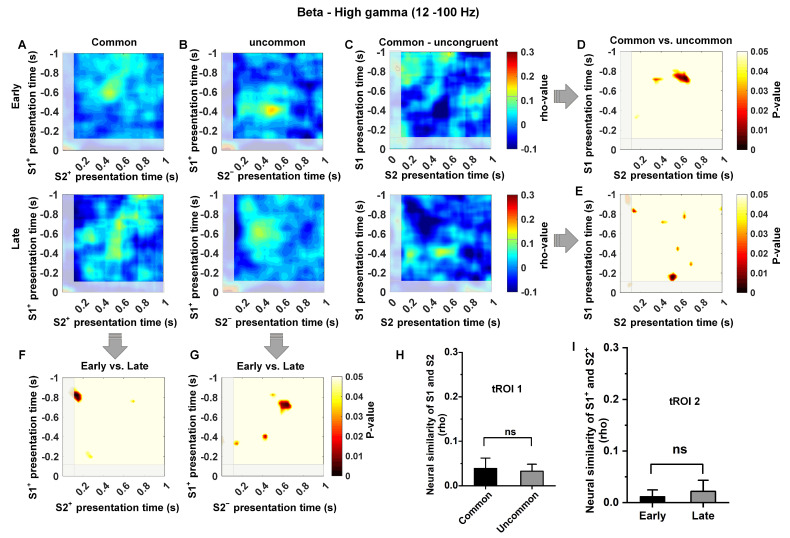
The quantification of neural similarity at the high-frequency band (12–100 Hz) for all conditions. Subfigure (**A**–**I**) corresponds to Figure 5, and shows a significant reduction in hippocampal neural similarity at the high-frequency (12–100 Hz) band compared to the low-frequency (1–12 Hz) band. There was no significant increase in neural similarity in the Hp for any of the conditions, including the early and late learning stages in tROI 1, or common and uncommon states in tROI 2. (n = 6; Early stage = 100 trials, Prob (S1^+^ choice) < 65%; Late stage = 100 trials, Prob (S1^+^ choice) > 90%; two sample test; “ns” indicates no significance).

**Table 1 animals-14-00431-t001:** Summary statistics of the different frequency band normalized power of S1^+^ choice trials in Step 1 in relation with $Q_{{S1}^{+}}$, and the power of S1^−^ choice trials in relation with $Q_{{S1}^{-}}$.

	Linear Regression of the Normalized Power $S1^{+}$ and $Q_{{S1}^{+}}$			$$\mathrm{Linear}\;\mathrm{Regression}\;\mathrm{of}\;\mathrm{the}\;\mathrm{Normalized}\;\mathrm{Power}\;\mathrm{of}\;S1^{-}\mathrm{and}\;Q_{{S1}^{-}}$$		
	R^2^	*p*-Values	F-Statistics	R^2^	*p*-Values	F-Statistics
1–12 Hz	0.0927	0.0503	6.0308	0.0537	0.0747	3.30
12–100 Hz	0.3554	<0.0001	32.23	<0.0001	0.9064	0.01

**Table 2 animals-14-00431-t002:** Summary statistics of the different frequency band normalized power of S2^+^ presentation trials in Step 2 in relation with $Q_{{S2}^{+}}$, and the power of S2^−^ presentation trials in relation with $Q_{{S2}^{-}}$.

	$\mathrm{Linear}\;\mathrm{Regression}\;\mathrm{of}\;\mathrm{the}\;\mathrm{Normalized}\;\mathrm{Power}\;\mathrm{of}\;S2^{+}\;\mathrm{and}\;Q_{{S2}^{+}}$			$\mathrm{Linear}\;\mathrm{Regression}\;\mathrm{of}\;\mathrm{the}\;\mathrm{Normalized}\;\mathrm{Power}\;\mathrm{of}\;S2^{-}\;\mathrm{and}\;Q_{{S2}^{-}}$		
	R^2^	*p*-Values	F-Statistics	R^2^	*p*-Values	F-Statistics
1–12 Hz	0.2186	0.0002	16.23	0.0018	0.7469	0.11
12–100 Hz	0.7551	<0.0001	158.30	0.3042	<0.0001	26.88

**Table 3 animals-14-00431-t003:** Decoding accuracy of pigeon’s behavior choice based on Hippocampal power distribution in different frequency bands.

Pigeon ID	Choice Decoding Accuracy	
	1–12 Hz	12–100 Hz
P090	0.752 ± 0.014	0.922 ± 0.012
P093	0.652 ± 0.019	0.815 ± 0.013
P014	0.521 ± 0.014	0.782 ± 0.016
P025	0.464 ± 0.016	0.778 ± 0.025
P021	0.651 ± 0.027	0.786 ± 0.020
P029	0.712 ± 0.020	0.801 ± 0.022
Average	0.626	0.814

## Data Availability

The datasets analyzed in the current study are available from the corresponding author on reasonable request.

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
