# Peer review of "The Hippocampus in Pigeons Contributes to the Model-Based Valuation and the Relationship between Temporal Context States"

_animals, 2024, doi:10.3390/ani14030431_

Round 1
Reviewer 1 Report
Comments and Suggestions for Authors
1) Summary
In this work entitled “The hippocampus in pigeons contributes to the model-based valuation and the relationship between temporal context states” of Yang et al., training pigeons to perform a two-step task and using a combination of neural analysis and computational modeling, try to expand the understanding of the hippocampus (HP) role in avian model-based learning showing how the pigeons use model-based inference to learn multi-step task. Analyzing in the HP of pigeons during a two-step learning task revealed that both low (1-12 Hz) and high-frequency (12-100 Hz) power distributions contribute to the decision-making process.
2) General concept comments.
Considering my area of specialization, I have no significant comments to make regarding the behavioral experimentation part and the type of tasks that the subjects must perform. However, regarding the part relating to the analysis of the electrophysiological signal, I have some requests to make to the authors.
Time-frequency analysis using Morlet wavelets is used as a method for estimating the power in the various frequency bands but the power profile (power spectral density - PSD) of these traces in the 1-100 Hz range is never shown. Would it be possible to have examples of these tracks, for the two Steps, accompanied by the PSD? Has an analysis of the power of the individual frequency bands been carried out? How do they vary between the two tasks, both in an absolute and relative sense?
In general, together with the request that is made later to add the depth coordinates and to specify the area of the HP that is reached, I believe that it would be very useful for the reader to integrate in the methods section, where the analysis is of the LFP signal, this data with an explanatory figure of the method, in order to better contextualize the data that is used.
3) Specific comments referring to line numbers, tables or figures that point out inaccuracies within the text or sentences that are unclear.
2.1.2. Subjects and Surgery: it is not said, or not well specified in the text, what depth is reached by the electrodes and from which area of the hippocampus the signal is extracted. I would ask the authors to add this information in this section.
2.1.3. Acquisition of neurophysiological data: specify at what sampling rate the signal is recorded. It is only then specified that the signal is resampled at 1 KHz.
2.1.4. Histological reconstructions: Would it be possible to see one of these histological reconstructions? Is it possible to add a figure with at least one of these reconstructions?
2.3.3. Neural similarity representation analysis of temporal context states
Row 268: Change “Delata (1-4 Hz)…” in “Delta”.
Author Response
Please see the attachment for details of the point-by-point response to the reviewer’s comments.

Reviewer 2 Report
Comments and Suggestions for Authors The authors present a fascinating study that examines the psychological correlates of a model-based learning model with the neurological correlates of the hypocampus in pigeons. In short, the study suggests that there is a link between these different correlates. Interestingly, the authors are able to demonstrate the distinct roles of low and high frequencies of hypocampal neural activity with the predictions of their model. Although my knowledge of model learning is a bit rusty - sorry, I am not really up to date - , I enjoyed reading this manuscript. The level of analysis is complex, but it's similar with what can be found in the literature for this type of research. In short, I generally found no major flaws in this manuscript. The study is well done, well described, the analysis is deep, and several alternative hypotheses are explored. Nevertheless, I would like to highlight a number of elements that the authors should address in the final version: I was a bit disappointed to see that model-based learning was limited to a two-step task. My understanding of model-based learning models is that they are used to explore the prediction of behavior as a function of environmental changes, and that this is done in a sequential manner over multiple stages (see introduction). Since the task used had only two stages, I seriously question the predictions of the model validated by the authors. In short, how will this model react if the number of steps to be predicted is increased from 2 to 4? I'd like the authors to address this question in their general discussion. Lines 76-95. I suggest deleting this paragraph. The authors summarize their methodology, results, and conclusions. In short, it kills the punch right out of the gate. Section 2.1.1 This section is written in present tense. Change it to past tense. Line 132. Specify that there are 2 sessions per day for a total of 60 sessions. Were days off (e.g., weekends)? Line 155 Did the pigeons lose weight before the trial started? We often reduce the weight of pigeons to 85% of their body weight before the start of the study to ensure their motivation. Line 221. I'd like a better explanation of the priors used by the authors. Why were these two types of priors chosen in relation to the estimated parameters? Line 233 Some studies. Please be more specific. Line 258. This sentence needs to be rephrased. Article 2.5 I was a little confused here. I had trouble understanding the distinction between regression and ANOVA (they're the same thing). Similarly, the distinction between parametric and non-parametric models in the results is not clear at all. What's more, since the same pigeons were tested repeatedly, repeated measures models should be used here. I'm thinking mixed models here, but I could be wrong as I didn't fully understand how the authors used regression models in this section. Figure 2 Can the authors explain the discrepancy between the results in Panel E and Panel F? Or am I missing something? Figure 2 Clearly there was a lot of variability between pigeons (Figure 2A, Figure 2B). Is there a way to illustrate this variability and include it in the analyses? By the way, I loved Figures 2 and 3. Table 1 I don't understand how the regression analyses were done. Sorry. Figure 4. How do you explain that most of the data is clustered at the right end of the spectrum? And that could affect your conclusion. And if we isolate the data from session 30 to session 60, there is clearly no relationship in the graphs. Can you explain that? Table 3 How do you explain the variability between pigeons for the hypicampal frequencies ranging from 1 to 12 Hz? General discussion The general discussion is limited. It focused primarily on the results, which were extensively presented (and sometimes repeatedly). Briefly, can the authors put their study in a more general context relative to the use of model-based learning and more deeply demonstrate how this particular study helps to extend the need for these kinds of learning models to understand behaviour and planning behavior. Comments on the Quality of English Language Section 2.1.1 This section is written in present tense. Change it to past tense.Author Response
Please see the attached document for the point-by-point response to the reviewer’s comments.
